# Creating small but meaningful representations of digital pathology images

**Editor:**

## Abstract

Representation learning is a popular application of deep learning where an object (e.g., an image) is converted into a lower-dimensional representation that still encodes relevant features of the original object. In digital pathology, however, this is more difficult because whole slide images (WSIs) are tiled before processing because they are too large to process at once. As a result, one WSI can be represented by thousands of representations - one for each tile. Common strategies to aggregate the "tile-level representations" to a "slide-level representation" rely on pooling operators or even attention networks, which all find some weighted average of the tile-level representations.

In this work, we propose a novel approach to aggregate tile-level representations into a single slide-level representation. Our method is based on clustering representations from individual tiles that originate from a large pool of WSIs. Each cluster can be seen as encoding a specific feature that might occur in a tile. Then, the final slide-level representation is a function of the proportional cluster membership of all tiles from one WSI. We demonstrate that we can represent WSIs in parsimonious representations and that these aggregated slide-level representations allow for both WSI classification and, reversely, similar image search.

**Keywords:** Digital pathology, representation learning, classification, similar image search

## 1. Introduction

Representation learning is a popular application of deep learning where an object (e.g., an image) is converted into a lower-dimensional representation that still encodes relevant features of the original object. This lower dimensional representation (also called embedding) can then be used in other machine learning tasks as a proxy for the original high-dimensional object. In digital pathology, the application of representation learning has to overcome an additional difficulty caused by the sheer size of whole slide images (WSIs). A typical WSI is so large that it cannot be processed as a whole. As such, WSIs are routinely divided into thousands of smaller parts, so-called tiles, that have reasonable dimensions ranging from a few hundred to a few thousand pixels squared. Representation learning is thus applied to a large number of tiles that constitute the WSI. As many problems in digital pathology relate to classifying a slide-level label (e.g., contains tumor, staging or presence of some biomarker), it is important to aggregate all the tile-level representations into a single representation of the WSI itself. This is the realm of multiple-instance learning and typical solutions include straightforward visualization of tile-level prediction and/or majority voting (Wei et al., 2019; Coudray et al., 2018; Pantanowitz et al., 2020), aggregation operators such as min/max/average pooling (Courtiol et al., 2018), as well as attention networks

(Tomita et al., 2018; Ilse et al., 2018; Shaban et al., 2020). Both the pooling operators as well as attention networks are prone to noise in the tile-level representations.

In this work, we propose a novel approach to aggregate the tile-level representations to create a meaningful slide-level representation. Our approach is based on first generating tile-level representations for tiles extracted from a large set of WSIs. Then we cluster the tile representations. Conceptually, each cluster represents a feature found in the tile (and hence in a WSI). Lastly, for each WSI we calculate the proportion of tiles that lies in any of the prior created clusters. This final step yields a representation of the WSI in only K numbers, where K is the number of clusters. We demonstrate that this aggregation works in conjunction with different deep-learning architectures to generate the initial tile-level representations. We refer to this final slide-level representation as a barcode for an associated WSI. The final representation can then be used with conventional machine learning algorithms to perform slide-level classifications and predictions. Moreover, we demonstrate that our barcode system can also be used to perform similar image search.

## 2. Methods

The overall methodology is illustrated in Figure 1. We start by extracting the tiles from WSIs and cleaning them (i.e., a quality control (QC) step, step 1). Then, we generate representations for the tiles (step 2) and cluster them using k-means clustering (step 3) to create clusters for salient features present in the tiles. Based on those clusters, we create a barcode for each WSI (step 4). Finally, and depending on the application, we train a classifier that takes the barcode and predicts a slide-level label (step 5). Below, those five steps are described in more detail and how they are used during training and inference.

### 2.1 Step 1: QC and tile extraction

While extracting the tiles from WSIs, we clean the data in an additional QC step (Step 1 in Figure 1). Overall, the QC step extracts tiles from the WSIs and then determines where the tissue is, whether the tile contains only adipose tissue, and whether the tissue is out-of-focus and/or covered by pen markers. To determine where the tissue is, we use a standard ResNet18 network, conceptually similar to the method used in Pantanowitz et al. (2020). We then pass the tiles through two other ResNet18 classifiers that label the tiles as in-focus (or not) and as pen markers (or not). We only keep tiles that are predicted to be in focus tissue without pen markers on them. This step is performed both during training for all WSIs in the training set and during inference for new WSIs.

### 2.2 Step 2: Generate tile-level representations

The next step is to generate tile-level representations (Step 2 in Figure 1). In principle, any deep learning network that generates a latent variable can be used for this task as long as the latent representation contains some meaningful information about the tiles. We have explored two approaches. First, we used a plain vanilla ResNet18 model that was trained using cross-entropy loss to perform tissue type detection. In this case, every tile gets the slide-level label during training. We used this setting to ensure that meaningful features related to tissue-type classification are captured by the tile-level representation. In a

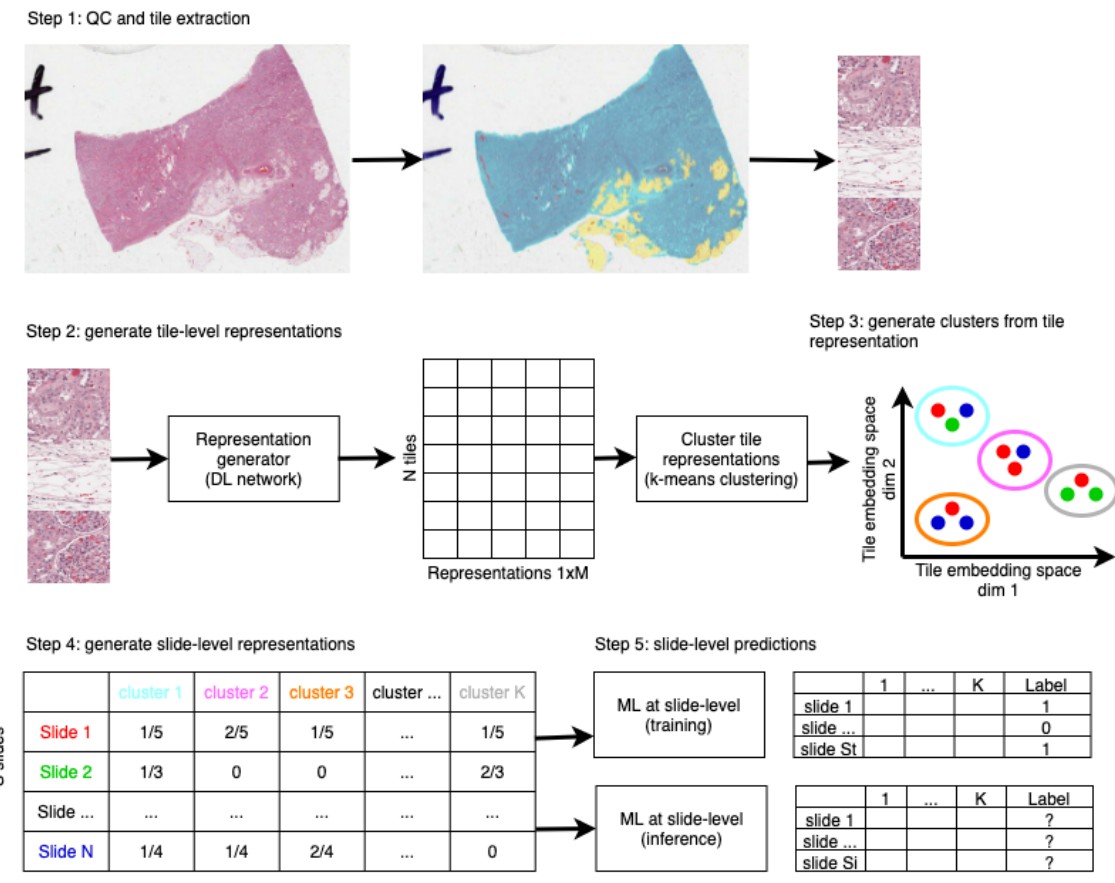

Figure 1: Schematic of novel approach. Step1: Perform QC to locate regions in the image that are in-focus and free of pen markers. Extract 224x224 tiles at 20x magnification from the WSI. Step 2: Generate a representation for each tile. Each tile yields a 1xM representation, where M depends on the network. For instance, the ResNet18 used in this work generates a 1x512 embedding. An NxM representation is generated from all N tiles extracted from all S WSIs where each representation is of length M. Step 3: All N tiles are clustered in K clusters (with k-means clustering in this work). In this schematic, we have three WSIs that have respectively 5, 4 and 3 tiles. The tiles from those WSIs are illustrated by a red, blue or green dot. In this schematic, K=4. Step 4: A barcode is generated for each WSI by calculating the proportion of tiles associated with that WSI that belong to each of the clusters. The rows in the left table constitute the final slide-level representation. Step 5: the barcodes are used to train a classifier to predict slide-level labels for the WSIs.

second approach, we used the recently proposed self-supervised "Bootstrap your own latent" (BYOL) method from Grill et al 2020. We used this approach to demonstrate that our

clustering-based aggregation method works as long as the tile-level representations capture some meaningful features from the tiles. The step to generate tile-level representations has to be performed both during training and inference.

### 2.3 Step 3 and 4: Aggregating tile-level to slide-level representation

The true novelty of our approach lies in the way we aggregate the many tile-level representations associated with a single WSI into a single, parsimonious slide-level representation (Step 3 and 4 in Figure 1). In this step, we take all tile-level representations and cluster them using k-means clustering. Conceptually, each cluster contains a salient feature found in the collection of tiles. The k-means model only needs to be created once during training. Once the clustering is done, we generate the final slide-level representation by checking the proportion of tiles from a WSI that belong to each of the clusters (Step 4 in Figure 1). This step yields one ultra-parsimonious $1 \times K$ representation for each WSI. This step needs to be performed during training and inference.

### 2.4 Step 5: Slide-level predictions

In the final step, we use the slide-level representations as a proxy for the WSI or its tiles and train a classifier to make predictions solely based on the $1 \times K$ representation. (Step 5 in Figure 1). In this work we used off-the-shelf XGBoost (Chen and Guestrin, 2016) to perform the classification.

We note that the slide-level representations are generic and are generated independent of the downstream classification tasks. Once we have calculated the representation for a given WSI, it can be used in different tasks without the need to perform any of the prior 4 steps.

## 3. Experiments

### 3.1 Data

To demonstrate the capabilities of our novel approach, we only used publicly available data obtained from The Cancer Genome Atlas (TCGA). We used TCGA data from 10 different tissues ('Bladder', 'Brain', 'Breast', 'Bronchus and lung', 'Connective, subcutaneous and other soft tissues', 'Kidney', 'Liver and intrahepatic bile ducts', 'Pancreas', 'Prostate gland', 'Thyroid gland'). We limited ourselves to the diagnostic slides prepared via formaldehyde fixation and paraffin embedding (FFPE) as they are generally of higher quality compared to fast-frozen (FF) slides. In total, this yielded 5768 WSIs. Training and test sets are stratified on the slide-level to avoid having tiles from one WSI in both training and test sets. Unless reported otherwise, we used 50 WSIs for each tissue type for testing, while the other WSIs were used for training.

First, we run our in-house artefact detection algorithm so we can exclude extracting tiles from the background, out-of-focus areas or areas covered by pen markers (See Step 1 in the Methods). We extracted 224x224 tiles at 5x and 20x magnification to test our approach at both levels of magnification. To create our final test set at 5x magnification, we randomly sampled 500 tiles from 50 randomly selected WSIs. The final training set comprised all other WSIs where the exact number of WSIs depended on the availability on TCGA (e.g.,

we used 1054 FFPE breast WSIs but only 124 FFPE WSIs of connective tissue). In case a WSI yielded less than 500 tiles, we used all tiles for that WSI.

To create the final test set at 20x, we randomly sampled 1000 tiles (when available) from 35 randomly selected WSIs. The training set was made up by randomly sampling 1000 tiles (or all tiles when fewer tiles were available) from 300 randomly selected WSIs (or the maximum number of available WSIs when fewer WSIs were available). We picked less WSIs at 20x for computational reasons. For both 5x and 20x, random subsampling of the tiles did not appear to be detrimental for the performance of downstream models (not shown).

## 3.2 Tissue type classification

In this work we focus on the task of tissue-type classification, which is good for benchmarking the novel approach. In this task the algorithm has to predict the tissue type. We performed several experiments.

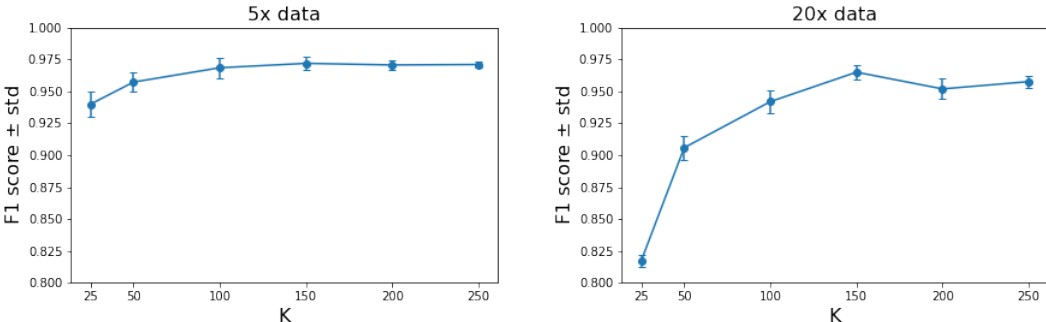

Figure 2:  Performance in the tissue type classification task using the novel slide-level representations as a function of the number of clusters (K). Performance using 5x data is high across a range of K. For 20x data K >= 150 yields the best results because at 20x the representations contain more numerous microscopic features compared to the macroscopic features captured at 5x magnification. The performance was assessed on the training set as that set allowed for 5-fold cross-validation. The error bars indicate the standard deviation for the different folds.

As a benchmark, we use a plain vanilla ResNet18 with cross-entropy loss to perform tile-level prediction. The tile-level prediction accuracy is 0.79 and 0.76 for 5x and 20x, respectively. F1 scores are identical because the test set is balanced with 50 WSIs for each tissue type. In this benchmark, the final slide-level prediction is calculated by majority voting and yields an accuracy of 0.92 and 0.89 for 5x and 20x, respectively. The slide-level accuracy is always better as noise in the tile-level predictions is leveled out by taking the majority vote.

To test our novel approach we used the aforementioned optimized ResNet18 network to generate tile-level representations. These tile-level representations were then passed through the pipeline to generate slide-level representations (Figure 1, Steps 3 and 4). Subsequently, we trained a classifier with XGBoost (Chen et al 2016) to predict the tissue types. Both

the k-means clustering as well as the XGBoost classifier were trained solely on the training data. To investigate the influence of K, we first assessed the performance on the training set by varying K. For each K in [25, 50, 100, 150, 200, 250] we performed 5-fold cross-validation. From Figure 2, we can see that the performance is better compared to the benchmark across a wide range of K. For 5x data, even small K yield good results, while good performance with 20x data is only obtained for $K \geq 150$. We assessed the influence of K on the training set as the test set is too small to perform a reliable 5-fold cross validation.

Next, we did the final assessment on the complete test set with K=150 where we achieved an F1 score of 0.91 and 0.86 for 5x and 20x data, respectively. As such, despite the enormous compression of the data (i.e., a WSI is summarized into a 1x150 vector), the F1 score is just marginally lower than the F1 score of the benchmark. This result validates our novel approach and demonstrates that our clustering-based aggregation method can capture (the) important features in a WSI.

To demonstrate that our method can extract and capture relevant features from any tile-level representations, we also used a ResNet18 network that was optimized using the self-supervised BYOL method. Note that this network was trained on different in-house, non-TCGA datasets originating from clinical trials. We generated the tile-level representations and aggregated the results with our clustering-based approach. Finally, we trained another XGBoost classifier and achieved a performance of 0.68 on our test set. Random guessing the tissue type yields an accuracy of 0.1 so even this network still performs reasonably well and validates that our approach can capture relevant features that are present in the tile-level embeddings (and see the Discussion).

### 3.3 Prediction of primary diagnosis

To demonstrate the generalizability of the clustering-based slide-level representations, we also performed another task, namely that of primary diagnosis prediction. For this task, we selected all brain WSIs we had in our TCGA dataset and predicted the tumor stage: lower-grade glioma (LGM) or glioblastoma (GBM). For this experiment, we used the slide-level representations generated earlier with the plain vanilla ResNet18 and the BYOL optimized ResNet18; both based on 20x tiles. We only optimized the classifier in Step 5 to predict LGM or GBM. With both initial networks to generate the tile-level representations, we achieved good accuracies of $\sim 0.8$ and $\sim 0.85$ depending on K, for BYOL and plain vanilla Resnet18, respectively (Figure 3). This validates that our slide-level representations capture information relevant for generic tasks in digital pathology.

### 3.4 Similar image search

Finally, we investigated whether our slide-level representations can be used in similar image search. In this experiment, we present one slide-level representation (i.e., the prototype) and then determine the nearest L neighbors in the K-dimensional representation space (with K=150). We used the full test set with 50 WSIs for each of the 10 tissue types and expect the nearest neighbors to be of the same tissue type as the prototype. This can be quantified by measuring the hit rate and the number of correct retrievals as a function of the number of nearest retrieved neighbors. The hit-rate starts off at 0.83 and converges to $\sim 0.95$ with 9 retrieved WSIs (Figure 4, left). The hit rate only shows the probability of hitting at least

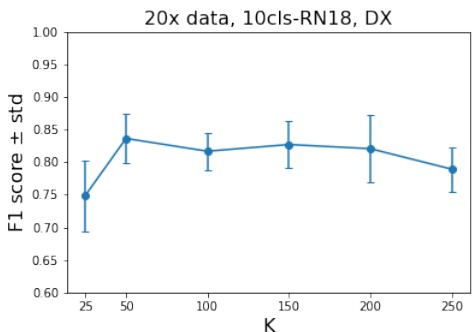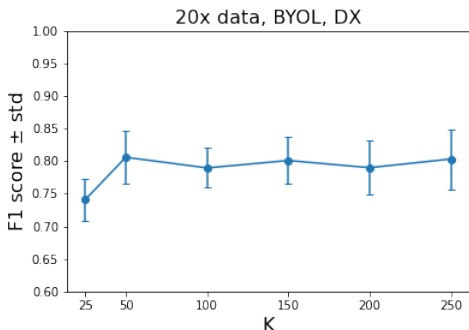

Figure 3: Performance in the primary diagnosis classification task using the novel slide-level representations as a function of the number of clusters (K). Both networks achieve similar F1 scores indicating that both capture generic relevant features that can be used for different tasks. The error bars represent the standard deviation in the 5-fold cross-validation.

one WSI of the same type. The number of correct retrievals are illustrated in Figure 4 (right) and shows the same trend: most of the retrieved nearest neighbors are in fact of the same type. These results show that our slide-level representations can also be used for rudimentary similar images search.

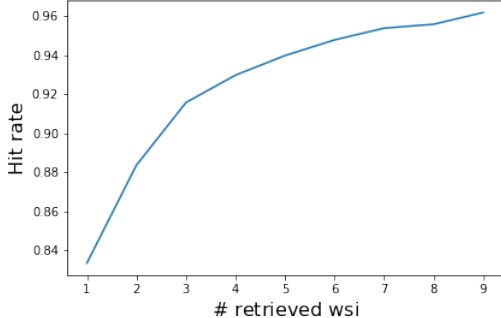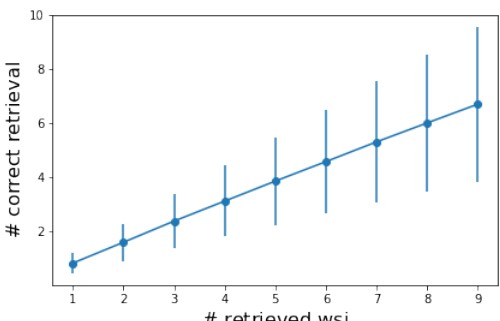

Figure 4: Similar image search based on the novel slide-level representations. Left: the probability of retrieving N nearest neighbors and having at least one of the same tissue type (the "hit rate"). Right: The number of correct retrieved images as a function of retrieved images.

## 3.5 Conclusion and discussion

We described a novel approach to generate slide-level representations for use in digital pathology. Rather than having to deal with thousands of representations associated with tiles extracted from a WSI, we aggregate those tile-level representations into meaningful slide-level representations. Due to the clustering of a large number of tile-level representa-

tions, we can produce extremely parsimonious representations of size $1 \times K$, where K can be as low as 100 (Figure 2 and 3) while still capturing generic WSI characteristics. Our results indicate that with those representations we can successfully perform tissue type type classification as well as primary diagnosis classification on the basis of the same representations.

We also expect that our approach can help open the black box of machine learning to some extent: conceptually, the clusters capture one (group) of spatial features that can be present in a WSI. As such, with low numbers of clusters, we can investigate what feature is present. Moreover, in combination with basic classifiers such as decision trees or logistic regression, we speculate that these models can indicate which features are related to slide-level labels. For instance, a WSI with a high proportion of tiles in a cluster representing adipose tissue can be related to breast. Similarly, a WSI with any tiles that fit in a cluster representing colloid would indicate thyroid tissue.

Generating slide-level representations associated with a certain magnification level provides a different view of the same WSI: at 5x magnification the features are of macroscopic nature while at 20x (or higher) the features represent microscopic features. Due to the extremely small size of the final embeddings, in the future we aim to combine (e.g., concatenate) slide-level representations at different magnifications to further improve classification tasks (He et al., 2015).

## 4. Acknowledgements

The results shown here are in part based upon data generated by the TCGA Research Network: https://www.cancer.gov/tcga.

## 5. Competing interests

The authors declare the following competing interests: P.A. and B.T-N. are employees of Roche and C.G. was employed by Roche at the time of this work. P.A. and B.T-N. have shares in the company.

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
