# OpenReview forum: "Creating small but meaningful representations of digital pathology images"
_MICCAI.org/2021/Workshop/COMPAY — COMPAY 2021_

### Official Review · Reviewer_tf8W · 2021-07-28
**A good paper with several open questions.**

**Rating:** 7
**Confidence:** 4

**Review:**

The authors describe a method to represent a WSI by its normalized histogram of clusters which are trained/found in a training set.
They claim that this histogram can be used as representative feature vector for the WSI for various classification tasks.

Although this is an interesting approach and surely interesting for the community, I do have several comments and see potential for improvements in the current manuscript. In general, I think that the proposed work could benefit from several obvious experiments to strengthen their proposed approach.

- It is unclear how much the clustering itself is contributing to the classification result. As a benchmark, you use a majority vote for the classified tiles of an image. But I assume a "fair" comparison would be to use the histogram over all classified tiles in a WSI instead of a majority vote. This histogram is the alternative to your cluster histogram.
- As a most simple representation of a tile, why don't you use an autoencoder, but create a classification task?
- It is not clear to me how the tile representation influences the classification. E.g., if you would use a primary Dx classification task to learn the tile representation, and then do the clustering, would the final primary prediction (section 3.3) improve? Or using auto-encoded tile representations or differently compressed tiles, would the performance of the final classifier change? How?
- Do you have any insight/knowledge of what the clusters represent?
- The authors may want to cite / compare to related work
https://arxiv.org/abs/2012.13955
https://arxiv.org/abs/1903.07013
https://arxiv.org/abs/2103.10626
The authors present results, but they do not compare to other state-of-the-art approaches. This makes it difficult to assess, if their approach is now the next big thing, or if it is just a different way to deal with WSI.

- The paper is nicely written and easy to understand.

---

### Official Review · Reviewer_G4V8 · 2021-08-06
**The authors of this paper introduce a method to create whole slide image-level representations by calculating the proportion of tiles in clusters.**

**Rating:** 5
**Confidence:** 3

**Review:**

The authors of this paper introduce a method to create whole slide image-level representations by calculating the proportion of tiles in clusters. They showed their representations can successfully classify tissue types in TCGA data, predict the brain tumor stage (lower-grade glioma vs glioblastroma), and retrieve whole slide images with the same tissue type. The authors attempt to solve an important problem in digital pathology which can lead to many applications such as tumor stage prediction and similar image search. The approach to create more compact representations is interesting. I have some concerns:
1. My understanding is that the main contribution of this work is to use clusters containing various features from tiles to create compact representations. The authors stated this multiple times in the paper, but the analysis is missing. This work would be complete if the authors show what kind of features those clusters are learning.
2. The authors need to describe Bootstrap your own latent (BYOL) method in detail in Section 2.2. In addition, is BYOL method better than a plain vanilla ResNet18 with cross-entropy loss? Section 3.2 seems to show that a classifier with XGBoost achieves F1 score of 0.91 but BYOL method achieves 0.68. Please describe Section 3.2 more clearly if my interpretation is incorrect.
3. In Figure 1, step 1, please indicate what blue and yellow regions are.

---

### Decision · Program_Chairs · 2021-08-25

Accept